# Bisphenol S Modulates Type 1 Diabetes Development in Non-Obese Diabetic (NOD) Mice with Diet- and Sex-Related Effects

**DOI:** 10.3390/toxics7020035

**Published:** 2019-06-23

**Authors:** Joella Xu, Guannan Huang, Tai L. Guo

**Affiliations:** 1Department of Veterinary Biosciences and Diagnostic Imaging, University of Georgia, Athens, GA 30602, USA; joella@uga.edu; 2Department of Environmental Health Sciences, University of Georgia, Athens, GA 30602, USA; guhuang@uga.edu

**Keywords:** Bisphenol S, type 1 diabetes, NOD mice, immunomodulation, glucose homeostasis, adult exposure, phytoestrogens

## Abstract

Bisphenol S (BPS) is a common replacement for bisphenol A (BPA) in plastics, which has resulted in widespread human exposure. Type 1 diabetes (T1D) is an autoimmune disease resulting from pancreatic β-cell destruction and has been increasing in incidence globally. Because of the similarities (e.g., endocrine disrupting) between BPS and BPA, and the fact that BPA was previously shown to accelerate T1D development in female non-obese diabetic (NOD) mice, it was hypothesized that BPS could contribute to the increasing T1D incidence by altering immunity with sex-biased responses. Adult female non-obese diabetic (NOD) mice were orally administered BPS at environmentally relevant doses (3, 30, 150 and 300 μg/kg), and males were given 0 or 300 μg/kg BPS. Females following 30 μg/kg BPS treatment on a soy-based diet had significantly delayed T1D development at the end of the study and decreased non-fasting blood glucose levels (BGLs) during the study. In contrast, BPS-exposed males on a soy-based diet showed an increased insulin resistance and varied BGLs. This might be a mixture effect with phytoestrogens, since males on a phytoestrogen-free diet showed improved glucose tolerance and decreased insulin resistance and CD25^+^ T cells. Additionally, while BPS altered BGLs in soy-based diet mice, minimal effects were observed concerning their immunotoxicity. Thus, BPS had sex- and diet-dependent effects on T1D and glucose homeostasis, which were likely caused by other mechanisms in addition to immunomodulation.

## 1. Introduction

Bisphenol S (BPS) is replacing bisphenol A (BPA), an established endocrine disrupting chemical (ED) used in a wide variety of polycarbonate plastics and epoxy resins. This has resulted in widespread exposure to BPS with its continuous buildup in the environment and use in plastic production [1,2]. It is of particular concern since, similar to BPA, recent studies have shown that BPS can act as an obesogen [3], alter macrophage cytokine production [4] and reduce zebrafish plasma insulin levels [5]. However, much remains unknown about BPS’s potential toxicities, including its effect on type 1 diabetes (T1D).

T1D, an autoimmune disease characterized by pancreatic β-cell destruction, has been increasing in incidence globally in both children and adults [6]. BPA has previously been shown by us and others to dysregulate adult glucose homeostasis in humans [7] and accelerate T1D development and/or autoimmune responses in female non-obese diabetic (NOD) mice exposed during adulthood or the juvenile period [8,9,10,11] as well as streptozotocin-treated C57BL/6 male mice [12]. Additionally, BPA has sex-biased effects including delaying T1D in male NOD mice compared to accelerating T1D in females [10]. 

The overall hypothesis of this study was that BPS would act similarly to BPA by altering T1D susceptibility in NOD mice through immunomodulation in a sex-dependent manner. Thus, we aimed to determine the sex-related effects of a relevant dose of BPS on T1D. Modulation of diabetes by EDs might depend on diet as we have previously found diet-dependent effects from exposure to genistein, an ED, on T1D in streptozotocin-induced diabetic B6C3F1 male mice and from perinatal BPA exposure on T1D development in NOD mice [11,13]. Therefore, an additional objective was to determine if diet-specific effects on immunity to modulate T1D development would be observed from BPS exposure in males. This is the first study to evaluate the effects of BPS on T1D, and in males, how dietary phytoestrogens may impact these effects.

## 2. Materials and Methods 

### 2.1. Animal Husbandry

Specific pathogen-free (SPF) NOD mice initially obtained from Taconic Biosciences (Hudson, NY) were bred in the Coverdell animal facility at University of Georgia (UGA). We chose NOD mice, a murine model that spontaneously develops T1D, since they have genetics and changes in immunity that lead to the destruction of pancreatic β-cells similar to humans and are an excellent model for assessing possible environmental factors for T1D risk [14]. Mice were kept in polysulfone cages with irradiated laboratory animal bedding and Bed-r’Nest for enrichment (The Andersons Inc., Maumee, Ohio) at 22–25 °C with relative humidity 50 ± 20 and a 12-h light/dark cycle. Negligible amounts of BPA have been reported to leach from new or used polysulfone cages maintained at room temperature [15,16]. Water and food were provided ad libitum. Mice had access to water through the animal facility’s automatic watering system. Animal caretakers were blinded to the exposure groups. In all experiments performed, animals were treated humanely with regard to alleviating animal suffering. An approved animal protocol (A2017 09-001-Y2-A2; Approved on: 2017-09-28) by the UGA Institutional Animal Care and Use Committee (IACUC) was followed for all procedures.

### 2.2. Bisphenol S (BPS) Exposure

Age-matched mice (8–15 wks old) were randomized into vehicle (VH) or BPS groups, and analysis of variance (ANOVA) was performed to ensure that background body weight (BW) and blood glucose levels (BGLs) were not significantly different among groups before treatment started. While this age range is after the pre-diabetic stage, it was chosen to measure the impact of BPS on T1D exacerbation as NOD mice typically develop overt diabetes starting between 10–14 wks old and continuing up to 30 wks old [17]. All mice were given corn oil via pipetting with either 0.05% ethanol (VH) or 0.05% ethanol that had BPS dissolved into it. Female mice were fed the soy-based PicoLab diet (LabDiet, St. Louis, MO, USA) that contains ~237–655 ppm isoflavone [18] and dosed daily with 0 (VH), 3, 30, 150 or 300 μg BPS/kg BW with 6 mice per group. Males were given 0 or 300 μg BPS/kg BW with 5–6 mice per group and fed with the PicoLab diet. Since soy phytoestrogens can have antagonistic or additive effects with BPA depending on the target cells [19,20], we also utilized the phytoestrogen-free 5K96 diet (TestDiet, St. Louis, MO, USA) in males with 10 mice per group to determine if there were any dietary interactions with BPS and phytoestrogens.

Compared with BPA, current human BPS exposure has been found at similar levels for occupational exposure, but at an order of magnitude lower for the general population [1]. Considering the levels of BPS exposure in relation to BPA, we chose this study’s doses based on (1) unconjugated serum BPA levels measured in mice given 400 μg/kg BPA were lower than those found in adult men and women [21], and (2) as a comparison to our previous studies on adult BPA exposure in NOD mice [10]. For males, we chose the 0 or 300 μg/kg doses to compare with our previous BPA findings as this dose also has relevance for human exposure. 

### 2.3. Body Weight, Blood Glucose Measurement and Diabetic Incidence

BW and non-fasting BGLs were measured every week. A Contour Blood Glucose Meter (Ascensia Diabetes Care, Parsippany, NJ, USA) was used to measure BGLs using a tail nick to collect a small sample of venous blood. Once 2 consecutive non-fasting BGL measurements were ≥250 mg/dL, mice were considered diabetic [13]. If BGLs were ≥600 mg/dL for 2 consecutive non-fasting BGL measurements or at the end of the study, mice were humanely euthanized using CO_2_ asphyxiation. Since female NOD mice developed diabetes faster, they needed to be euthanized sooner than the males. While diabetes incidence was not as high as reported for NOD females (e.g., 90%) at the time of euthanizing, the rate of diabetes development was consistent with our other studies in female NOD mice [10,18]. None of the males developed diabetes during the studies, and all males were euthanized at the end of the study. Therefore, diabetes incidence could not be compared.

### 2.4. Glucose and Insulin Tolerance Tests

For the glucose tolerance test (GTT), mice were fasted overnight (15 h), had their fasting BW and BGLs measured, and then injected i.p. with 2 g/kg BW glucose (Sigma; [22]). For the insulin tolerance test (ITT), baseline BW and BGLs were measured, and then mice were injected i.p. with 1.5 IU/kg BW insulin (Sigma; [23]). BGLs were measured at 15, 30, 60 and 120 min after glucose or insulin injection.

### 2.5. Organ Collection and Flow Cytometry

Spleen, pancreas, kidneys with adrenals, liver and thymus were collected at euthanization, cleaned of connective tissue, and weighed. Spleens were mashed in 3 mL phosphate-buffered saline (PBS) solution on ice. Flow cytometric analysis was performed for quantifying leukocyte populations with different combinations of fluorochrome-labeled antibodies (diluted 1:80; BD PharMingen, San Diego, CA, USA) for all mice, including cluster of differentiation (CD) 40L-B220 (PE-FITC), CD5-CD24 (PE-FITC) and CD44-CD40 (PE-FITC) along with isotype matched irrelevant antibodies for controls. In addition, the combination of CD4-CD8-CD25-Mac3-CD45R (V450-APCH7-APCA-FITC-PE) was used for females, and CD4-CD8-CD25 (V450-APCH7-APCA) was used for the males. The combination of CD18-Mac3 (PE-FITC) was also used for phytoestrogen-free diet males. After antibody addition, cells were incubated in the dark for 30 min at 4 °C, and then washed and enumerated with a Becton Dickinson LSRII Flow Cytometer (BD Biosciences, San Jose, CA, USA) in which log fluorescence intensity was read. Red blood cells were eliminated by using a high forward scatter threshold, and each sample had at least 5000 cells counted. Analysis was done using FlowJo Version 10 software (FlowJo LLC, Ashland, Oregon, USA).

### 2.6. Statistical Analysis

The rate of diabetes development and total diabetes incidence over time were analyzed with Likelihood ratio and Logrank test, respectively. Correlational analysis was conducted using Spearman’s correlation test. For all other data sets, Dunnett’s test (VH as the reference group) was used for parametric data and the Wilcoxon test for non-parametric data, which was determined by unequal variances analysis (Bartlett’s test). A group was considered statistically significant if *p* < 0.05. JMP Pro 13 (SAS Inc., Cary, NC, USA) and GraphPad Prism 7 (GraphPad Software Inc., La Jolla, CA, USA) were used for statistical analysis and data visualization.

## 3. Results

### 3.1. BPS Exposure Had Protective Effects on Type 1 Diabetes (T1D) In Females on A Soy-Based Diet 

To determine the effect of BPS on T1D, female NOD mice were exposed to BPS at doses of 0, 3, 30, 150 or 300 μg/kg during adulthood. No significant effect was observed for the total incidence of diabetes (Logrank test was used for total diabetes incidence and Likelihood ratio for rate of diabetes development), but diabetes incidence in the 30 μg/kg BPS exposed mice was significantly lower on day 85 following the first dose (Figure 1A). Additionally, BPS at the two lowest doses (3 and 30 μg/kg) decreased non-fasting BGLs on day 29 (Figure 1B) and the highest dose (300 μg/kg BPS) decreased fasting BGLs before glucose challenge (0 min) in the 1 month (day 38) GTT (Figure 1C). However, the 2-month (day 66) GTT (Figure 1D) and ITTs at 1 month (35 days; Figure 1E) and 2 months (63 days; Figure 1F) were not significantly altered. 

Flow cytometric analysis suggested that %splenic CD4^–^CD25^+^ and CD8^–^CD25^+^ cells was increased following the 150 μg/kg BPS exposure (Table 1). In addition, the number of CD8^+^CD25^+^ cells was significantly increased at the 300 μg/kg BPS dose (data not shown). No significant differences were observed for either the percentages or the number of other T cell populations, macrophages, B cell populations or CD40^+^ antigen-presenting cells (APCs; Table 1 and data not shown). However, when the expression levels of surface markers by various cell populations were evaluated, the mean fluorescence intensity (MFI) of CD8 by CD8^+^CD4^–^ cells was significantly increased, while that of CD40 by CD40^+^CD44^–^ cells was significantly decreased at the 300 μg/kg BPS dose (Figure 2A,B). Additionally, CD40 MFI by CD40^+^CD44^–^ cells positively correlated with BGLs on day 29 (0.4343; *p* = 0.0384) and CD8 MFI by CD8^+^CD4^–^ cells positively correlated with BGLs on day 56 (0.5741; *p* = 0.0042). BW showed no significant differences throughout most of the study periods when compared to the VH, except for day 77 when mice given 300 μg/kg BPS had a significantly reduced BW (Figure 2C). No significant effect was found on organ weights (data not shown).

### 3.2. BPS Exposure Varied Blood Glucose Levels (BGLs) in Males on a Soy-Based Diet 

Next, to ascertain if there was a sex-biased effect on T1D from BPS exposure, we exposed NOD males during adulthood. Because the findings from female studies were mostly negative, one dose of 300 μg/kg was selected to compare with our previous findings with BPA exposure in males [10], and this dose might provide levels in serum closer to what was found in humans based on BPA [21]. No significant effect was found for the 1-month (day 28; Figure 3A) or 2-month (day 60) GTTs (Figure 3B). However, the BGLs were significantly increased in the 1 month (day 32) ITT at 30 and 60 min after challenge (Figure 3C), suggesting an increased insulin resistance. ITTs at 2 months (day 62; Figure 3D) and 5 months (day 147; Figure 3E) did not show significant differences between the groups. For the BGL time course, BPS exposure significantly increased non-fasting BGLs on days 6 and 13, and then, decreased them on day 75 when compared to the VH (Figure 3F). Immune cell populations, including splenic B and T cells, were not significantly altered when the data were expressed as percent values (Table 2), while the numbers of CD40^+^CD44^–^, CD40^+^CD44^+^, CD40^-^CD44^+^, CD4^+^CD25^−^, CD4^+^CD25^+^, and CD4^-^CD8^+^ cells were significantly decreased by BPS (data not shown). When the expression levels of surface markers by various cell populations were evaluated, the MFI of CD40 by CD40^+^CD44^+^ cells and that of CD24 by CD24^+^CD5^−^ cells were significantly increased by BPS (Figure 4A,B). However, BGLs did not significantly correlate with the MFI of CD40 by CD40^+^CD44^+^ cells or that of CD24 by CD24^+^CD5^−^ cells (data not shown). BW and organ weights were not significantly altered (Figure 4C and data not shown). 

### 3.3. BPS Exposure Improved Glucose Tolerance and Insulin Sensitivity in Males on a Phytoestrogen-Free Diet 

Since previous studies have shown that phytoestrogens can attenuate BPA’s effects [24,25], we treated NOD males with BPS while on a phytoestrogen-free diet for 70 days. The 1 month GTT (day 27; Figure 5A) and ITT (day 32; Figure 5B) were not significantly altered; however, the 2 month GTT (days 67; Figure 5C) showed a reduced BGL at the 15 min time point. The ITT at day 61 (Figure 5D) also had reduced BGLs at the 60 min timepoint from BPS exposure. After 70 days of BPS exposure on the phytoestrogen-free diet, the mice were switched to the soy-based diet for the remaining 31 days of the study to determine the effect of dietary phytoestrogens. The GTT at day 102 had reduced fasting BGLs at the 0 min time point (Figure 5E). The ITT at day 98 (Figure 5F) had reduced BGLs at the 60 min timepoint from BPS exposure. For the BGL time course, BPS significantly reduced non-fasting BGLs on days 5 and 87 after the first dose (Figure 5G). No significant effects were found in BW (Figure 5H). Flow cytometric analysis showed that BPS decreased the percentages of CD4^+^CD25^+^, CD8^+^CD25^+^, CD4^–^CD25^+^, and CD8^–^CD25^+^ cells, and the numbers of CD4^+^CD25^+^, CD4^–^CD25^+^, and CD8^–^CD25^+^ cells (Table 3 and data not shown). No significant effects were found in other immune cell populations (Table 3). While the spleen weight was not significantly altered, the absolute and percent thymus weights were significantly decreased, and the percent kidneys were increased compared to the VH (Table 4).

## 4. Discussion

This is the first study to determine whether BPS can alter T1D risk and glucose homeostasis. In female mice, the effects on diabetes, BGLs (non-fasting and fasting) and immunity were found to be dose-related. EDs including BPS have been shown to have a non-monotonic dose response [26,27], which may be the reason for the dose-related effects we observed. Of particular importance, 30 μg/kg BPS had a trend of protecting female mice from T1D development with a significant effect found on days 29 and 85 for BGLs and percent diabetic, respectively. However, immunity seemed not to be significantly altered except for the MFI of CD8 by CD8^+^CD4^–^ and that of CD40 by CD40^+^CD44^–^ cells in the 300 μg/kg BPS group and %splenic CD4^-^CD25^+^ and CD8^-^CD25^+^ T cells in the 150 μg/kg BPS group. Although the 150 μg/kg BPS group did not show any alterations of T1D from the increased CD25^+^ T cells, CD25^+^ T cells may not have a primary role in regulating T1D. Rather the sensitivity of the effector T cells to regulation by CD25^+^ T cells seemed to be of more importance for T1D development in NOD mice [28]. The positive correlation of CD8 MFI and CD40 MFI with BGLs along with their significant alterations in the 300 μg/kg BPS group suggests BPS might exert effects on BGLs through altering expression levels of these cell markers. However, whether and how the changes in cell marker levels can impact BGLs still needs to be determined. It is possible that an alteration of these cell marker expression levels could affect cell functionality, such as decreased antigen interaction and T cell signaling with a lower CD8 expression [29]. The lack of effect on immunity from the 30 μg/kg BPS exposure may be due to the low numbers (*N* = 3) of VH mice at the end of the study or other mechanisms such as altering cytokine levels, gut microbiota and/or epigenetics, which should be further investigated in future studies with increased numbers of mice in each group. Taken together, like BPA, our BPS data further supports that exposure from lower doses of BPS has greater impacts on certain endpoints (e.g. diabetes incidence) than that of higher BPS doses, which is consistent with a non-monotonic dose response, possibly due to receptor competition, receptor down-regulation, competing monotonic curves for different endpoints or other unclarified mechanisms [30]. In addition, these results were also in contrast to what we and others previously found with BPA exposure, which accelerated T1D and/or increased pro-inflammatory immune factors in NOD females [8,9,10,31], suggesting that BPS might act through different mechanisms than BPA.

In males on the soy-based diet, BPS exposure impaired glucose homeostasis initially and increased insulin resistance in the 2-month ITT. Since this effect did not continue throughout exposure, it is likely that the body compensates for the impact of BPS exposure on BGLs allowing for homeostasis to be reached over a relatively short time. Similar to our results, BPS was recently found to impair glucose homeostasis in male zebrafish through alteration of mRNA levels relevant to glucose metabolism [5]. Although the biphasic effect on BGLs indicated that BPS may possibly have a protective effect if exposure had continued longer than 5 months, there is a need for additional studies on how long-term BPS exposure impacts T1D. Our current findings of lower BGLs on day 75 only without an improved glucose tolerance or a decreased insulin resistance are not sufficient to conclude a protective effect from long-term exposure. In addition, similar to the females exposed to BPS in our studies, variation of BGLs and insulin resistance in males following BPS exposure did not coincide with alterations in immune cell populations. These results were also in contrast with our previous study with BPA where exposure in NOD males had clear protective effects on T1D and an anti-inflammatory shift [10]. This further suggests that BPS works through different mechanisms from BPA.

Unlike the soy-based diet males, BPS showed protective effects in the phytoestrogen-free diet males including reduced BGLs, improved glucose tolerance and increased insulin sensitivity. Furthermore, switching these mice to a soy-based diet on day 70 failed to attenuate the protective effects as seen from the reduced fasting BGLs in the GTT and insulin resistance in the ITT after 3 months of BPS exposure. Conversely, the immune cell populations consisting of CD25^+^ cells were decreased by BPS exposure, which is more consistent with exacerbation of T1D in NOD mice [32]. These results, along with the results discussed above, lacked a consistent shift towards pro-inflammation or anti-inflammation in immune cell populations of NOD mice exposed to BPS, and thus point to other mechanisms contributing to BPS’s alteration of BGLs. Future studies examining gut microbiota, cytokines and epigenetic factors will help elucidate the mechanisms of BPS modulation of glucose homeostasis. Additionally, the sex-dependent effects from BPS exposure should be further explored such as determining the effects of BPS on hormone levels.

It has been shown that dietary phytoestrogens from soy, such as genistein, can have protective effects on T1D [13,18,33,34]. Additionally, previous studies have shown that phytoestrogens can attenuate BPA’s effects [11,24,25]. Although determining the mechanisms of how dietary phytoestrogens interact with BPS was not the main focus in these studies, the underlying mechanisms for the interactions might include competing for the same receptor and/or one or more ED impacting cell cycle and altering the effects of other ED(s) exposed to. However, the exact mechanism(s) of mixture effects still remain relatively unknown [35]. In our studies with male NOD mice that were switched from a phytoestrogen-free diet to a soy-based diet, a similar pattern of decreased glucose levels was observed for both diets, which was in contrast to the early increases in non-fasting BGLs when male NOD mice were kept on soy-based diet from the beginning. Additional experiments to determine mechanisms of mixture effects by focusing on BPS and genistein are warranted.

This study is limited by the fact that we did not have the full range of doses given to males as we did in females. Similarly, males were given two different diets, while females were only studied with the soy-based diet. Future studies using males and females exposed to a range of BPS doses and on different diets are needed for a better comparison between the sexes along with a larger group size. In our studies, none of the male NOD mice developed diabetes after over 21 wks of age, so our conclusions in males can only be used for BPS modulation of glucose homeostasis to suggest potential effects on T1D risk. While it was possible that diabetes might have developed in some animals if we had continued the treatment for a longer period (e.g., up to 30 wks of age), these results in males still provided useful information about the implications of BPS exposure. Future studies are needed using male mice that have a higher incidence of T1D such as excluded flora (EF) NOD mice or Akita mice. 

In conclusion, BPS exposure had dose-related protective effects on T1D in females. As for males, treatment with 300 μg/kg BPS impacted glucose homeostasis that was differentially altered based on diet. The males on a phytoestrogen-free diet had consistently protective effects on BGLs, while the males on the soy-based diet had some adverse effects initially, which suggested that phytoestrogens likely had a mixture effect with BPS. These results on T1D and BGLs were different from what was found in previous studies with BPA. Unlike BPA, minimal effects were found on immunity when mice were exposed to BPS on a soy-based diet. This suggests that BPS uses different mechanisms from BPA to alter glucose homeostasis and T1D. Thus, the impact of BPS on T1D and possible mechanisms such as gut microbiota and epigenetics should be further evaluated.

## Figures and Tables

**Figure 1 toxics-07-00035-f001:**
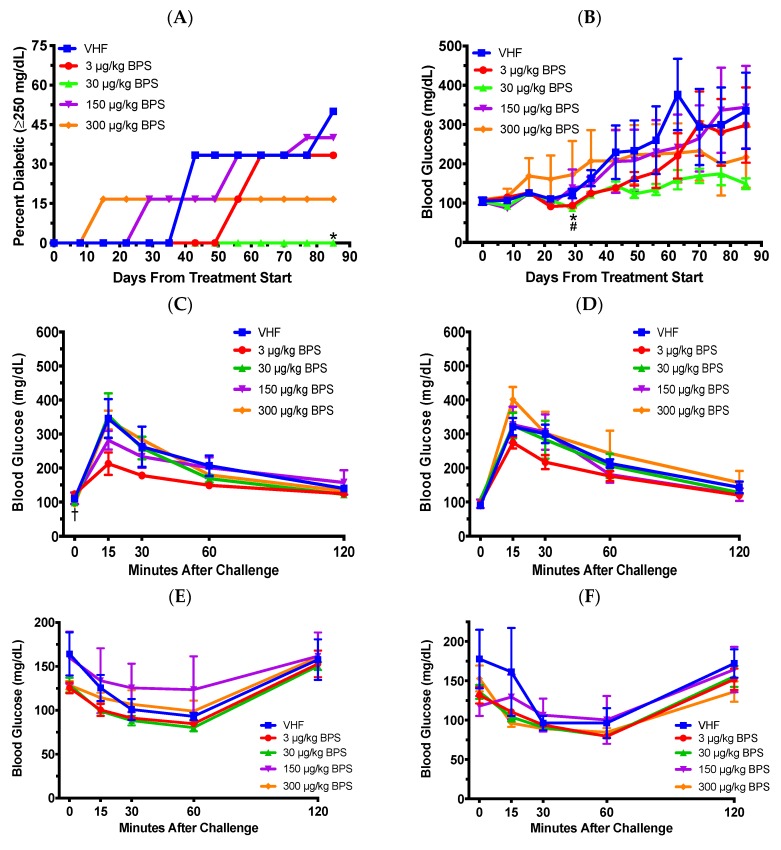
Diabetic incidence, blood glucose levels (BGLs), glucose tolerance tests (GTT) and insulin tolerance tests (ITT) in female non-obese diabetic (NOD) mice on the soy-based diet. (**A**) Type 1 diabetes (T1D) incidence (*N* = 5–6). Blood glucose ≥250 mg/dL was considered diabetic. (**B**) Time course of non-fasting BGLs. Also shown are GTTs after 1 month (**C**) and 2 months (**D**) of BPS exposure (*N* = 5–6), and ITTs after 1 month (**E**; *N* = 5–6) and 2 month (**F**; *N* = 3–5) of BPS exposure. The values are presented as mean ± SEM. *, *p* < 0.05 for 30 μg/kg BPS vs. vehicle (VHF) control group. ^#^, *p* < 0.05 for 3 μg/kg BPS vs. VHF group. ^†^, *p* < 0.05 for 300 μg/kg BPS vs. VHF group.

**Figure 2 toxics-07-00035-f002:**
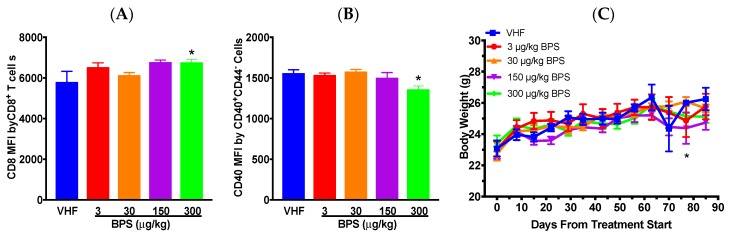
Flow cytometric analysis and body weight in female NOD mice on the soy-based diet. The expression of CD8 by CD4^−^CD8^+^ cells (**A**) and of CD40 by CD40^+^CD44^–^ cells (**B**). *N* = 3 for the VHF and 150 μg/kg dose groups; *N* = 6 for the 3 and 30 μg/kg dose groups; *N* = 5 for the 300 μg/kg dose group. (**C**) Changes in body weight over time (*N* = 4–6). The values are presented as mean ± SEM. *, *p* < 0.05 for 300 μg/kg BPS vs. VHF group.

**Figure 3 toxics-07-00035-f003:**
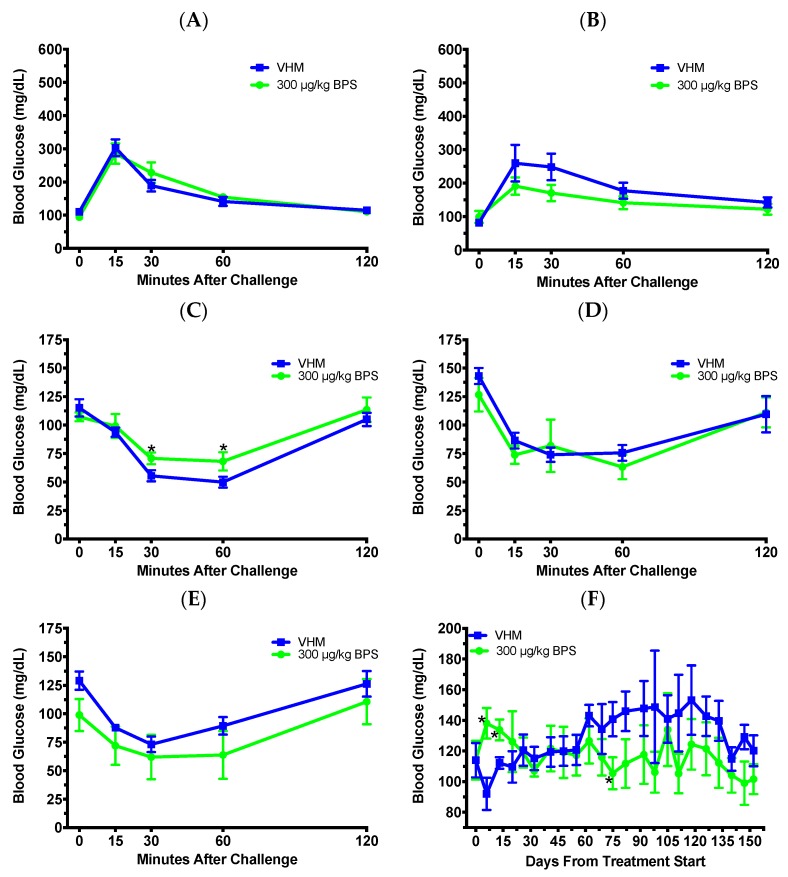
Tolerance tests and blood glucose levels (BGLs) in male NOD mice on the soy-based diet. GTT after 1 month (**A**) and 2 months (**B**) of BPS exposure, and ITT after 1 month (**C**), 2 months (**D**) and 5 months (**E**) of BPS exposure are shown. (**F**) Time course of non-fasting BGLs. The values are presented as mean ± SEM. *, *p* < 0.05. *N* = 5–6. VHM, vehicle males.

**Figure 4 toxics-07-00035-f004:**
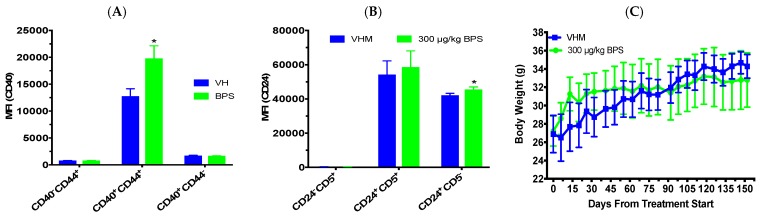
Immune cell populations and body weight in male NOD mice on the soy based diet. (**A**) The expression of CD40 by CD40^+^CD44^+^ cells. (**B**) The expression of CD24 by CD24^+^CD5^-^ cells. (**C**) Changes in body weight over time. The values are presented as mean ± SEM. *, *p* < 0.05. *N* = 5–6. VHM, vehicle males.

**Figure 5 toxics-07-00035-f005:**
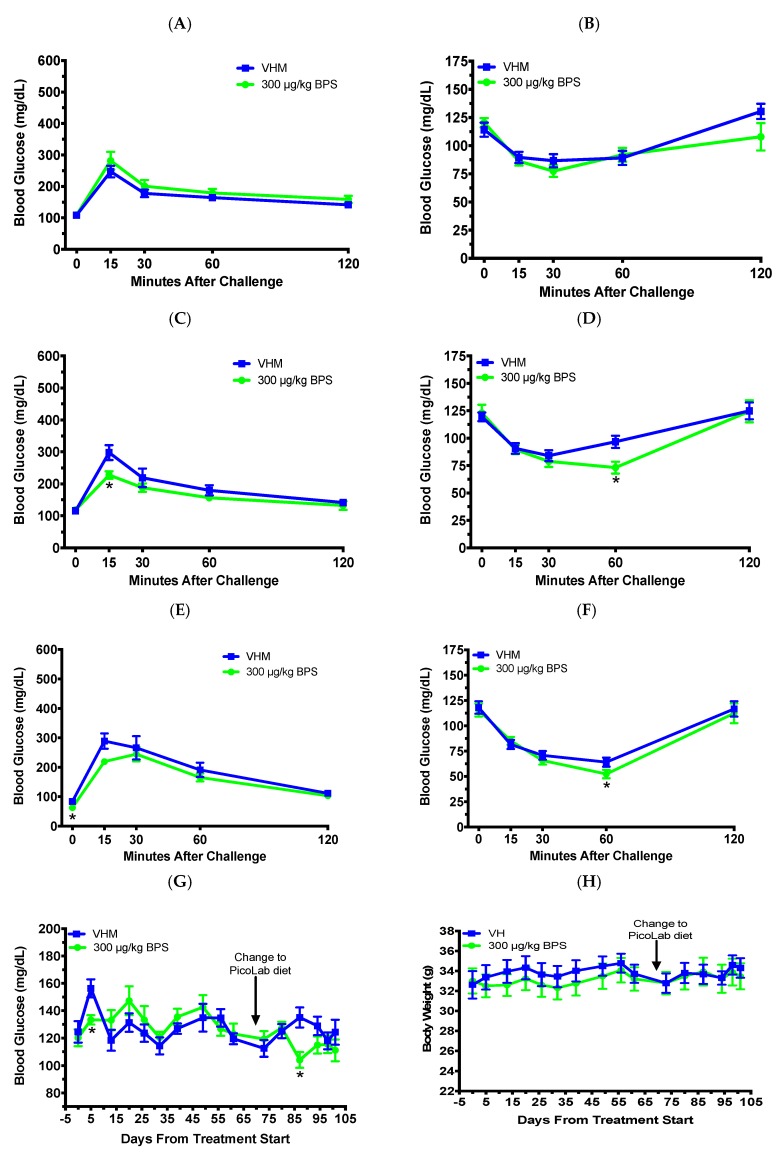
Tolerance tests, BGLs and body weight in male NOD mice initially on a phytoestrogen-free diet for 70 days and then switched to the soy-based diet for another 31 days. Glucose tolerance test (GTT) and insulin tolerance test (ITT) were conducted after BPS exposure for one month (**A**: GTT; **B**: ITT), 2 months (**C**: GTT; **D**: ITT) and 3 months (**E**: GTT; **F**: ITT). (**G**) Time course of non-fasting BGLs. (**H**) Changes in body weight over time. The values are presented as mean ± SEM. *, *p* < 0.05. *N* = 7–10. VHM, vehicle males.

**Table 1 toxics-07-00035-t001:** Percentages of splenic immune cell populations in female NOD mice on the soy-based diet.

Antibody Combinations	Group	+/− (%)	+/+ (%)	−/+ (%)
CD3/CD45R	VHF	9.87 ± 0.89	0.81 ± 0.16	36.93 ± 5.66
	3	10.72 ± 1.91	0.93 ± 0.12	35.78 ± 2.17
	30	9.18 ± 0.42	0.86 ± 0.06	34.27 ± 1.29
	150	12.04 ± 1.58	1.17 ± 0.17	37.40 ± 1.93
	300	9.96 ± 0.59	1.18 ± 0.11	36.24 ± 1.36
CD4/CD8	VHF	7.44 ± 1.17	1.82 ± 0.10	9.26 ± 1.29
	3	7.70 ± 1.31	1.62 ± 0.27	8.39 ± 0.83
	30	6.38 ± 0.47	1.94 ± 0.13	10.54 ± 0.87
	150	8.53 ± 1.17	2.39 ± 0.11	11.47 ± 0.37
	300	6.82 ± 0.34	2.42 ± 0.20	12.64 ± 1.46
CD4/CD25	VHF	7.68 ± 0.98	0.97 ± 0.12	0.36 ± 0.07
	3	7.73 ± 1.33	1.11 ± 0.12	0.48 ± 0.05
	30	6.83 ± 0.43	0.91 ± 0.09	0.40 ± 0.03
	150	8.87 ± 1.22	1.36 ± 0.02	0.61 ± 0.09 *
	300	7.63 ± 0.46	1.00 ± 0.93	0.50 ± 0.04
CD8/CD25	VHF	5.71 ± 0.61	0.37 ± 0.10	0.85 ± 0.11
	3	5.66 ± 0.80	0.39 ± 0.05	1.06 ± 0.11
	30	6.85 ± 0.62	0.32 ± 0.03	0.87 ± 0.08
	150	8.14 ± 0.44	0.49 ± 0.05	1.36 ± 0.09 *
	300	8.78 ± 1.03	0.49 ± 0.05	0.86 ± 0.06
CD24/CD5	VHF	67.40 ± 1.23	4.50 ± 0.03	16.47 ± 0.73
	3	63.17 ± 2.60	4.18 ± 0.31	19.77 ± 3.02
	30	66.95 ± 1.05	4.35 ± 0.36	16.08 ± 0.61
	150	63.37 ± 2.05	4.36 ± 0.35	19.50 ± 2.85
	300	66.42 ± 1.50	4.47 ± 0.24	16.32 ± 1.09
CD40/CD44	VHF	32.77 ± 2.35	3.83 ± 0.64	2.61 ± 0.59
	3	33.50 ± 2.68	3.89 ± 0.45	2.61 ± 0.28
	30	30.32 ± 1.58	3.67 ± 0.27	3.00 ± 0.13
	150	34.23 ± 2.95	3.59 ± 0.87	2.51 ± 0.55
	300	32.30 ± 1.60	2.61 ± 0.24	2.34 ± 0.20
B220/CD40L	VHF	28.53 ± 5.97	0.59 ± 0.11	1.66 ± 0.16
	3	33.10 ± 2.83	0.59 ± 0.08	1.62 ± 0.20
	30	29.97 ± 2.67	0.53 ± 0.09	1.83 ± 0.35
	150	34.10 ± 3.91	0.58 ± 0.11	1.48 ± 0.21
	300	29.52 ± 1.91	0.42 ± 0.07	1.21 ± 0.15
Mac3/CD45R	VHF	5.57 ± 0.50	7.15 ± 1.29	29.33 ± 4.63
	3	5.57 ± 0.27	8.32 ± 0.68	27.27 ± 1.76
	30	5.70 ± 0.32	7.94 ± 0.88	25.87 ± 0.70
	150	6.44 ± 0.67	11.10 ± 0.71	26.13 ± 1.47
	300	5.65 ± 0.67	10.32 ± 1.13	25.58 ± 0.76

* *p* < 0.05 as compared to the vehicle female (VHF) control group. The values are presented as mean ± standard error of the mean (SEM). *N* = 3 for the VHF and 150 μg/kg dose groups; *N* = 6 for the 3 and 30 μg/kg dose groups; *N* = 5 for the 300 μg/kg dose group.

**Table 2 toxics-07-00035-t002:** Percentages of splenic immune cell populations in male NOD mice on the soy-based diet.

Antibody Combinations	Group	+/− (%)	+/+ (%)	-/+ (%)
CD4/CD8	VHM	10.03 ± 1.27	2.03 ± 0.51	13.02 ± 0.67
	300	9.60 ± 0.44	1.56 ± 0.31	12.94 ± 1.45
CD4/CD25	VHM	10.56 ± 0.81	1.19 ± 0.09	1.24 ± 0.29
	300	9.85 ± 0.65	1.03 ± 0.10	1.12 ± 0.30
CD8/CD25	VHM	8.44 ± 0.48	0.60 ± 0.11	1.67 ± 0.19
	300	8.89 ± 1.73	0.53 ± 0.08	1.47 ± 0.32
CD24/CD5	VHM	66.40 ± 1.93	2.17 ± 0.10	23.00 ± 1.56
	300	65.94 ± 2.33	2.25 ± 0.39	24.44 ± 2.37
CD40/CD44	VHM	18.74 ± 0.89	2.20 ± 0.27	3.12 ± 0.21
	300	16.83 ± 1.43	1.77 ± 0.15	2.95 ± 0.26
B220/CD40L	VHM	15.64 ±1.25	0.86 ±0.08	3.14±0.15
	300	12.09 ±1.58	0.71 ±0.11	3.13±0.26

* *p* < 0.05 as compared to the vehicle male (VHM) control group. The values are presented as mean ± SEM. *, *p* < 0.05. *N* = 4–5. VHM, vehicle males.

**Table 3 toxics-07-00035-t003:** Percentages of splenic immune cell populations in male NOD mice initially on a phytoestrogen-free diet for 70 days and then switched to the soy-based diet for another 31 days.

Antibody Combinations	Group	+/− (%)	+/+ (%)	-/+ (%)
CD4/CD8	VHM	10.17 ± 0.68	1.23 ± 0.12	10.48 ± 0.55
	300	7.83 ± 0.91	2.08 ± 0.66	10.02 ± 1.29
CD4/CD25	VHM	9.46 ± 0.78	1.62 ± 0.12	3.30 ± 0.48
	300	8.61 ± 0.60	0.94 ± 0.19 *	1.49 ± 0.31 *
CD8/CD25	VHM	5.92 ± 0.40	1.14 ± 0.10	3.68 ± 0.48
	300	6.28 ± 0.90	0.80 ± 0.08 *	1.57 ± 0.44 *
CD24/CD5	VHM	71.91 ± 1.63	1.88 ± 0.06	19.44 ± 1.30
	300	73.37 ± 1.26	1.57 ± 0.13	17.73 ± 1.15
CD40/CD44	VHM	13.80 ± 0.69	2.10 ± 0.15	3.65 ± 0.29
	300	12.07 ± 0.50	1.70 ± 0.16	3.91 ± 0.64
B220/CD40L	VHM	12.21 ± 0.84	0.71 ± 0.05	2.60 ± 0.20
	300	10.87 ± 0.72	0.71 ± 0.05	2.28 ± 0.13
MAC-3/CD18	VHM	1.92 ± 0.23	2.49 ± 0.27	1.88 ± 0.19
	300	1.67 ± 0.09	2.15 ± 0.16	1.71 ± 0.32

* *p* < 0.05 as compared to the vehicle male (VHM) control group. The values are presented as mean ± SEM. *, *p* < 0.05. *N* = 7–10. VHM, vehicle males.

**Table 4 toxics-07-00035-t004:** Organ weights in male NOD mice initially on a phytoestrogen-free diet for 70 days and then switched to the soy-based diet for another 31 days.

**Absolute Organ Weights (mg)**					
Treatment	Spleen	Pancreas	Kidneys	Liver	Thymus
Vehicle	84.0 ± 3.5	236.5 ± 12.3	487.8 ± 12.8	1750.7 ± 59.8	28.5 ± 1.7
300 μg/kg BPS	80.0 ± 6.0	219.2 ± 12.8	494.8 ± 20.5	1696.8 ± 70.3	19.1 ± 1.4 *
**Organ to Body Weight (%)**					
Treatment	%Spleen	%Pancreas	%Kidneys	%Liver	%Thymus
Vehicle	0.25 ± 0.01	0.69 ± 0.03	1.42 ± 0.02	5.10 ± 0.07	0.08 ± 0.01
300 μg/kg BPS	0.24 ± 0.01	0.66 ± 0.03	1.50 ± 0.02 *	5.07 ± 0.08	0.06 ± 0.00 *

Data are presented as mean ± SEM. *, *p* < 0.05 using Student’s t-test or Wilcoxon based on whether equal variance assumption was met as compared to the respective vehicle control group. *N* = 9–10.

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
