# Peer review of "Bisphenol S Modulates Type 1 Diabetes Development in Non-Obese Diabetic (NOD) Mice with Diet- and Sex-Related Effects"

_toxics, 2019, doi:10.3390/toxics7020035_

Round 1

Reviewer 1 Report

This article investigates the role of BPS on T1D incidence in non-obese diabetic mice. Although the scientific question is of great interest, the present study presents some major limitation due to the study design. Moreover substantial modifications need to be made to improve the manuscript.  

The author test five BPS doses, namely 0, 3, 30, 150, and 300 μg BPS/kg BW in females NOD mice feed on soy-based diet, but then test only two doses (0 and 300 μg BPS/kg BW) in males but repeat the experiment in two groups: one fed with a soya-based diet and the other with phytoestrogen‐free diet. The results obtained for females and for males are finally not comparable. The authors should further explain this choice and acknowledge this as a main limitation of the study.

Another main limitation is related to the number of animal per group that is very limited. The authors should at least refer to this as a main limitation when they interpret the results in the discussion.

Finally, the hypothesis and the objectives are not clearly stated in the introduction. The authors say that the hypothesis is that “BPS would act similar to BPA by altering T1D susceptibility depending on sex through immunomodulation in NOD mice.” but do not mention the impact of the diet. Plus they stress that the effect could depend from the sex but then they tested different thigs in males and females.  

Overall I think the results of this study have a scientific interest but the introduction, the methodologic section and the discussion have to be modified so to be more transparent highlighting the limitations in order to put the reader in the position of interpreting correctly the results. 

Introduction

In the introduction the main and secondary objectives of the study should be clearly stated because how it is written now is confusing. I suggest the author to put the hypothesis underlying the study at the end of the introduction and specify the objectives.

Moreover at line 46 the authors justify the choice of NOD mice but since they haven’t described yet the methods used in the study this sentence is out of place. Or they generalize the fact the advantages of using NOD mice or they move the sentence in the methods.

The modulation of ED effects depending on the diet should be further presented and the underlying biological mechanisms explained.

Materials and methods

It is not clear why males were given 0 or 300 μg BPS/kg BW and not also 3, 30, 150 μg BPS/kg BW like the females. This needs to be explained especially because as the authors say in the discussion “BPS have been shown to have a non‐monotonic dose response”.

Overall it is not clearly stated how many animals where in each group: 6 female mice per group and 10 male mice? Add a sentence that clearly presents the numbers.

The rationale behind the choice of the doses is confusing. Sentence line 78-80 has to be rewritten end better explained.

Why the authors tested only males on phytoestrogen‐free diet and not also females?

Results

Line 121 “of 0‐300 μg/kg during adulthood. » is confusing because it seems the authors used only doses when instead they used four.  

Discussion

A paragraph highlighting strengths and limitations of the study is needed.

The first sentence of the conclusion is misleading since the authors have tested only one dose in males they cannot compare with the results they obtained in females: “In conclusion, BPS exposure had dose‐related protective effects on T1D in females, while in males, glucose homeostasis was differentially altered based on diet. »

Author Response

We would like to thank the reviewers for these insightful comments on the manuscript. We have made changes to the manuscript according to the reviewers’ comments, and we believe these changes have greatly improved the manuscript. Detailed responses can be found below in blue font. If you need more information, please feel free to contact us.

Reviewer 1

Comments and Suggestions for Authors

This article investigates the role of BPS on T1D incidence in non-obese diabetic mice. Although the scientific question is of great interest, the present study presents some major limitation due to the study design. Moreover substantial modifications need to be made to improve the manuscript.  

The author test five BPS doses, namely 0, 3, 30, 150, and 300 μg BPS/kg BW in females NOD mice feed on soy-based diet, but then test only two doses (0 and 300 μg BPS/kg BW) in males but repeat the experiment in two groups: one fed with a soya-based diet and the other with phytoestrogenfree diet. The results obtained for females and for males are finally not comparable. The authors should further explain this choice and acknowledge this as a main limitation of the study.

Another main limitation is related to the number of animals per group that is very limited. The authors should at least refer to this as a main limitation when they interpret the results in the discussion.

These limitations have been added to the discussion.

Finally, the hypothesis and the objectives are not clearly stated in the introduction. The authors say that the hypothesis is that “BPS would act similar to BPA by altering T1D susceptibility depending on sex through immunomodulation in NOD mice.” but do not mention the impact of the diet. Plus they stress that the effect could depend from the sex but then they tested different things in males and females.  

We have moved the hypothesis to the end of the introduction and further clarified our objectives. As both females and males were tested at the 0 and 300 μg/kg doses, we left did not remove the sex-dependency from our overall hypothesis, since we believe this dose is useful for comparing sex differences.

Overall I think the results of this study have a scientific interest but the introduction, the methodologic section and the discussion have to be modified so to be more transparent highlighting the limitations in order to put the reader in the position of interpreting correctly the results. 

Introduction

In the introduction the main and secondary objectives of the study should be clearly stated because how it is written now is confusing. I suggest the author to put the hypothesis underlying the study at the end of the introduction and specify the objectives.

We have moved the hypothesis to the end of the introduction and further clarified our objectives. As both females and males were tested at the 0 and 300 μg/kg doses, we left did not remove the sex-dependency from our overall hypothesis, since we believe this dose is useful for comparing sex differences.

Moreover at line 46 the authors justify the choice of NOD mice but since they haven’t described yet the methods used in the study this sentence is out of place. Or they generalize the fact the advantages of using NOD mice or they move the sentence in the methods.

This has been moved to 2.1 of the methods section.

The modulation of ED effects depending on the diet should be further presented and the underlying biological mechanisms explained.

This paragraph has been moved to the discussion and expanded upon.

Materials and methods

It is not clear why males were given 0 or 300 μg BPS/kg BW and not also 3, 30, 150 μg BPS/kg BW like the females. This needs to be explained especially because as the authors say in the discussion “BPS have been shown to have a nonmonotonic dose response”.

We have added this as a limitation of our study to the discussion.

Overall it is not clearly stated how many animals where in each group: 6 female mice per group and 10 male mice? Add a sentence that clearly presents the numbers.

We have modified section 2.2 to more clearly state the amount of mice per group for each study.

The rationale behind the choice of the doses is confusing. Sentence line 78-80 has to be rewritten end better explained.

This section has been reworded.

Why the authors tested only males on phytoestrogenfree diet and not also females?

We have added this as a limitation of our study in the discussion.

Results

Line 121 “of 0300 μg/kg during adulthood. » is confusing because it seems the authors used only doses when instead they used four.  

We have modified this sentence.

Discussion

A paragraph highlighting strengths and limitations of the study is needed.

These limitations have been added to the discussion.

The first sentence of the conclusion is misleading since the authors have tested only one dose in males they cannot compare with the results they obtained in females: “In conclusion, BPS exposure had doserelated protective effects on T1D in females, while in males, glucose homeostasis was differentially altered based on diet. »

This sentence has been modified to provide a clearer understanding of our paper’s conclusion.

Reviewer 2 Report

This is an important study showing that exposure to BPS has different effects on autoimmune diabetes in NOD mice compared to BPA, reducing diabetes in female mice and having sex and diet-dependent effects on glucose homeostasis.

Comments:

1) Could the authors please add the information about blinding of caretakers to exposure groups and blinding of investigators for GTT, ITT and blood glucose measurement?

2) Could the autors please comment on the choice of feed -phytoestrogencontaining- to NOD mice in relation to diabetes development, since phytoestrogen is protective against diabetes.

3) Starting the exposure at the age of 8-15 weeks could already be after normal diabetes development starts (early signs of insulitis), could the autors please comment on this?

4) The insidence of diabetes is rather low, normaly about 90% of the female NOD mice develop diabetes, could you please comment on this? Were the animals stressed by the gavage treatment?

5) Could the authors please elaborate on the statement on page 11 line 227: how could BPS affect BGLs through altering expression levels of cell markers?

6) There was no diabetes developed in the male mice, but was the study continued long enough for this to be noted? How long was the study conducted?

7) The protective effect of BPS on diabetes development in the female mice was only seen in the 30ug/kg BPS exposure, while only the higher doses resulted in effects on immune cell populations, do you suggest in the line 229 page 11 that the BPS exposure was too low to see an effect? 300ug/kg seems to have an effect on diabetes insidence, but not the 150 dose. Is a non-monotonic dose effect expected with different mechanisms at higher doses? Effects on cellpopulations by the 150 dose does not seem to affect the diabetes insidence, could you elaborate on this? Dop you have enough power for the insidence?

8) The long term exposure of male mice to BPS seem to lower the glucose Levels (significantly at 75 day only), this would be considered a long term protective effect of BPA as seen on the phytoestrogenfree diet day 87?

Minor comments:

1) Figure 1 is missing the "A" and "B" in the figure.

2) Figure 1 Quality would be improved by having the same labeling also in panel A (3 and 30 change colours and symbols to have a uniform layout With the other panels).

3) Table 4: Should treatment be 300mikrog/kg BPS (not 30)?

4) page 11, line 237; (wording) it is likely that the body compensates…

5) page 11, line 255 decreased by BPS..., which is more

Author Response

We would like to thank the reviewers for these insightful comments on the manuscript. We have made changes to the manuscript according to the reviewers’ comments, and we believe these changes have greatly improved the manuscript. Detailed responses can be found below in blue font. If you need more information, please feel free to contact us.

Comments and Suggestions for Authors

This is an important study showing that exposure to BPS has different effects on autoimmune diabetes in NOD mice compared to BPA, reducing diabetes in female mice and having sex and diet-dependent effects on glucose homeostasis.

Comments:

1) Could the authors please add the information about blinding of caretakers to exposure groups and blinding of investigators for GTT, ITT and blood glucose measurement?

I have added the information about blinding of the animal caretakers. However, the GTT, ITT and blood glucose measurements were measured without being blinded to the groups, so this information was not added in.

2) Could the authors please comment on the choice of feed -phytoestrogen containing- to NOD mice in relation to diabetes development, since phytoestrogen is protective against diabetes.

We have added an explanation for the choice of a phytoestrogen-free diet.

3) Starting the exposure at the age of 8-15 weeks could already be after normal diabetes development starts (early signs of insulitis), could the authors please comment on this?

We have added an explanation to the adult NOD methods section.  NOD mice typically develop overt diabetes at 10-14 wks old (70-98 days), but can become diabetic until 30 weeks (~210 days) of age.

4) The incidence of diabetes is rather low, normally about 90% of the female NOD mice develop diabetes, could you please comment on this? Were the animals stressed by the gavage treatment?

We have added a comment in the methods section 2.3.

5) Could the authors please elaborate on the statement on page 11 line 227: how could BPS affect BGLs through altering expression levels of cell markers?

We have elaborated on this statement.

6) There was no diabetes developed in the male mice, but was the study continued long enough for this to be noted? How long was the study conducted?

We have clarified on this in the discussion.

7) The protective effect of BPS on diabetes development in the female mice was only seen in the 30ug/kg BPS exposure, while only the higher doses resulted in effects on immune cell populations, do you suggest in the line 229 page 11 that the BPS exposure was too low to see an effect? 300ug/kg seems to have an effect on diabetes incidence, but not the 150 dose. Is a non-monotonic dose effect expected with different mechanisms at higher doses? Effects on cell populations by the 150 dose does not seem to affect the diabetes incidence, could you elaborate on this? Do you have enough power for the incidence?

More details have been added to this part of the discussion to address these questions.

8) The long term exposure of male mice to BPS seem to lower the glucose Levels (significantly at 75 day only; Figure 3F), this would be considered a long term protective effect of BPS as seen on the phytoestrogen free diet day 87 (Figure 5G)?

The discussion for the soy-based diet males has been expanded to address this question.

Minor comments:

1) Figure 1 is missing the "A" and "B" in the figure.

This has been corrected, and it was due to formatting.

2) Figure 1 Quality would be improved by having the same labeling also in panel A (3 and 30 change colours and symbols to have a uniform layout with the other panels).

This has been corrected.

3) Table 4: Should treatment be 300mikrog/kg BPS (not 30)?

This has been corrected.

4) page 11, line 237; (wording) it is likely that the body compensates…

This has been corrected.

5) page 11, line 255 decreased by BPS..., which is more

This has been corrected.

Round 2

Reviewer 1 Report

I feel that the quality and clearness of the manuscript has greatly improved after the revision made by the authors. There are some limitations concerning the study design but now they are clearly stated and taken into account in the discussion and conclusion.  

Author Response

Dear Reviewer,

We would like to thank you for all the helpful comments on the manuscript. For this round of revision, we have fixed any grammar errors and clarified the sentence on page 11 (lines 257-259).

Reviewer 2 Report

The Authors have now answered my comments and improoved the manuscript. Still, there are some few minor comments I would appriciate to get clarified.

1) reduced binding (to what?, B-cells, T-cells, antigens, please clarify)

2) Page 11, line 252: (rewrite to:) ...support that effects from lower doses of BPS have grater impact on certain endpoints compared to higher doses, in line with a non-monotonic dose response, due to yet unclear mechanisms (or something that sounds slightly better than the present line).

3) It has been shown that genistein also have protective effects on diabetes development in NOD females by elevating insulin levels and decreasing glucose Levels (Choi et al. 2008 Diabetes Metab Res Rev 2008 Jan-Feb; 24(1):74-81). Please add this information and reference.

4) page 11, line 264 ...indicatet that BPA

5)page 12, line 312: This suggests that BPS

Author Response

Dear Reviewer,

We would like to thank you for all the helpful comments on the manuscript. We have made changes to the manuscript according to the reviewer's comments, and we believe these changes have greatly improved the manuscript. For this round of revision, we have fixed any grammar errors and clarified the sentence on page 11 (lines 257-259). Additionally, we have included the citation about genistein on page 12 (lines 297-299). Below is a point-by-point response to the comments.

1) reduced binding (to what?, B-cells, T-cells, antigens, please clarify)

This sentence has been modified for clarification.

2) Page 11, line 252: (rewrite to:) ...support that effects from lower doses of BPS have greater impact on certain endpoints compared to higher doses, in line with a non-monotonic dose response, due to yet unclear mechanisms (or something that sounds slightly better than the present line).

This sentence has been rewritten for clarification.

3) It has been shown that genistein also have protective effects on diabetes development in NOD females by elevating insulin levels and decreasing glucose Levels (Choi et al. 2008 Diabetes Metab Res Rev 2008 Jan-Feb; 24(1):74-81). Please add this information and reference.

We have added this information and citation in our manuscript.

4) page 11, line 264 ...indicated that BPA

We have fixed this sentence.

5)page 12, line 312: This suggests that BPS

We have fixed this sentence.